# Why do biting horseflies prefer warmer hosts? tabanids can escape easier from warmer targets

Gábor Horváth[1]*, Ádám Pereszlényi[1,2], Ádám Egri[3,4], Tímea Tóth[1], Imre Miklós Jánosi[5]

1 Department of Biological Physics, ELTE Eötvös Loránd University, Budapest, Hungary, 2 Department of Zoology, Hungarian Natural History Museum, Budapest, Hungary, 3 MTA Centre for Ecological Research, Danube Research Institute, Budapest, Hungary, 4 Evolutionary Systems Research Group, MTA Centre for Ecological Research, Tihany, Hungary, 5 Department of Physics of Complex Systems, ELTE Eötvös Loránd University, Budapest, Hungary

* gh@arago.elte.hu

## Abstract

Blood-sucking horseflies (tabanids) prefer warmer (sunlit, darker) host animals and generally attack them in sunshine, the reason for which was unknown until now. Recently, it was hypothesized that blood-seeking female tabanids prefer elevated temperatures, because their wing muscles are quicker and their nervous system functions better at a warmer body temperature brought about by warmer microclimate, and thus they can more successfully avoid the host's parasite-repelling reactions by prompt takeoffs. To test this hypothesis, we studied in field experiments the success rate of escape reactions of tabanids that landed on black targets as a function of the target temperature, and measured the surface temperature of differently coloured horses with thermography. We found that the escape success of tabanids decreased with decreasing target temperature, that is escape success is driven by temperature. Our results explain the behaviour of biting horseflies that they prefer warmer hosts against colder ones. Since in sunshine the darker the host the warmer its body surface, our results also explain why horseflies prefer sunlit dark (brown, black) hosts against bright (beige, white) ones, and why these parasites attack their hosts usually in sunshine, rather than under shaded conditions.

## Introduction

Blood-sucking horseflies (tabanids) prefer warmer (sunlit, darker) host animals against colder (shaded, brighter) ones and generally attack them in sunshine [1, 2, 3, 4, 5]. Tabanids attack black cattle more frequently than white ones [6]. Among white, brown and black cattle, black individuals are the preferred targets of *Tabanus* spp. horsefly attacks [7]. The attractiveness of sunlit brown horses to tabanids is about four times larger than that of sunlit white ones, and in comparison with a white horse, a brown horse spends two times longer in a tabanid-free shaded forest than in a sunny field with intense tabanid attacks [1]. The most effective tabanid traps use shiny black decoys [8, 9, 10, 11, 12, 13, 14, 15]. The so-called H-traps (composed of a

**Data Availability Statement:** All relevant data are within the manuscript and its Supporting Information file.

**Funding:** This work was supported by the grant NKFIH K-123930 (Experimental Study of the

Functions of Zebra Stripes: A New Thermophysiological Explanation) received by Gábor Horváth from the Hungarian National Research, Development and Innovation Office. Ádám Egri was supported by the Economic Development and Innovation Operational Programme (GINOP-2.3.2-15-2016-00057), the grant NKFIH PD-131738 and the János Bolyai Research Scholarship of the Hungarian Academy of Sciences. The funders had no role in study design, data collection and analysis, decision to publish, or preparation of the manuscript.

**Competing interests:** The authors have declared that no competing interests exist.

bright tent with a shiny black sphere suspended below it) placed in sunny sites capture significantly more female tabanids than at shaded sites [16]. The reason for this is that sunlit shiny dark targets reflect light at the Brewster's angle with higher degrees of linear polarization $d$ than shaded ones [17, 18], and host-seeking female tabanids prefer high $d$-values independent of the direction of polarization [19]. Thus, shiny black decoys used to catch horseflies work due to their colour and reflected degree of polarization, rather than their temperature.

After these experimental and observational findings concerning tabanid thermal preference, Horváth *et al.* [20] showed that *Tabanus tergestinus* horseflies prefer sunlit warm shiny black targets over sunlit or shaded cold ones with the same optical characteristics. Furthermore, they hypothesized that a blood-seeking female tabanid prefers elevated temperatures, because her wing muscles are quicker and her nervous system functions better in a warmer microclimate, and thus she can more successfully avoid the host's parasite-repelling reactions by prompt take-offs. Of course, there could also be other reasons why blood-sucking horseflies might prefer to attack warmer host animals. For example, to increase sweating, the capillaries could be enlarged near the epidermis of warmer hosts, which could be advantageous for blood-sucking insects.

The prediction of the hypothesis of Horváth *et al.* [20] is that the escape success of horseflies that land on host animals increases with increasing surface temperature. To test this prediction, we studied the escape success of tabanids that landed on black targets as a function of the surface temperature, and measured the coat temperature of differently coloured sunlit and shaded horses with thermography. The results of our field experiments presented here corroborated prediction which explains why blood-seeking horseflies prefer sunlit dark (warmer) host animals.

## Results

As expected, the surface temperature $T$ of the sunlit back of horses decreased in the colour order black > brown > beige > white, and the mean temperature $<T>$ of the bellies had a smaller standard deviation $\Delta T$ than the backs (Fig 1, S1–S4 Figs). The minimum and maximum surface temperatures of horses were: black: 30.9–54.6˚C, brown: 31.2–44.6˚C, beige: 32.6–46.2˚C, white: 31.0–46.6˚C. The range $T_{max}$—$T_{min}$ and $\Delta T$ increased with increasing $<T>$ (Fig 2, S1–S4 Tables).

Fig 3 displays the surface temperature range $T_{min} \leq T \leq T_{max}$ of barrels and the proportions of escape success and capture rate of tabanids that landed on the barrels under different illumination and thermal conditions. Considering experiments 1–3, the escape success was the highest on the sunlit air-filled barrel (85.4%, $\chi^2 = 48.167$, df = 1, p < 0.001), it was the lowest on the shaded water-filled barrel (28%, $\chi^2 = 4.84$, df = 1, p = 0.02781), and on the shaded air-filled barrel it was in between the former two (54.5%, $\chi^2 = 0.36364$, df = 1, p = 0.5465). Under sunlit conditions in experiment 4, the escape success on the air-filled barrel (81.3%) was significantly higher by a factor of 2.3 ($\chi^2 = 34.9634$, df = 1, p < 0.001) than that on the water-filled barrel (35.6%). In experiment 5, tabanids could escape also with a significantly higher success ($\chi^2 = 32.5403$, df = 1, p < 0.001) from the sunlit side of the air-filled barrel (86.4%) than from its shaded side (39.5%), similarly to the sunlit (45%) and shaded (29.4%) sides of the water-filled barrel ($\chi^2 = 3.1832$, df = 1, p = 0.074398). The numbers of captured and escaped tabanids were not significantly different in the following situations: shaded side of the air-filled barrel in experiment 2, shaded side of the air-filled barrel in experiment 5, and sunlit side of the water-filled barrel in experiment 5 (Fig 3, S5 Table). As illustrated in Fig 3 (S6–S10 Tables), the surface of air-filled barrels was always warmer than that of water-filled ones, and the sunlit surface of a given barrel was warmer than its shaded side. All these results support our hypothesis that tabanids can escape more successfully from warmer targets than from cooler ones.

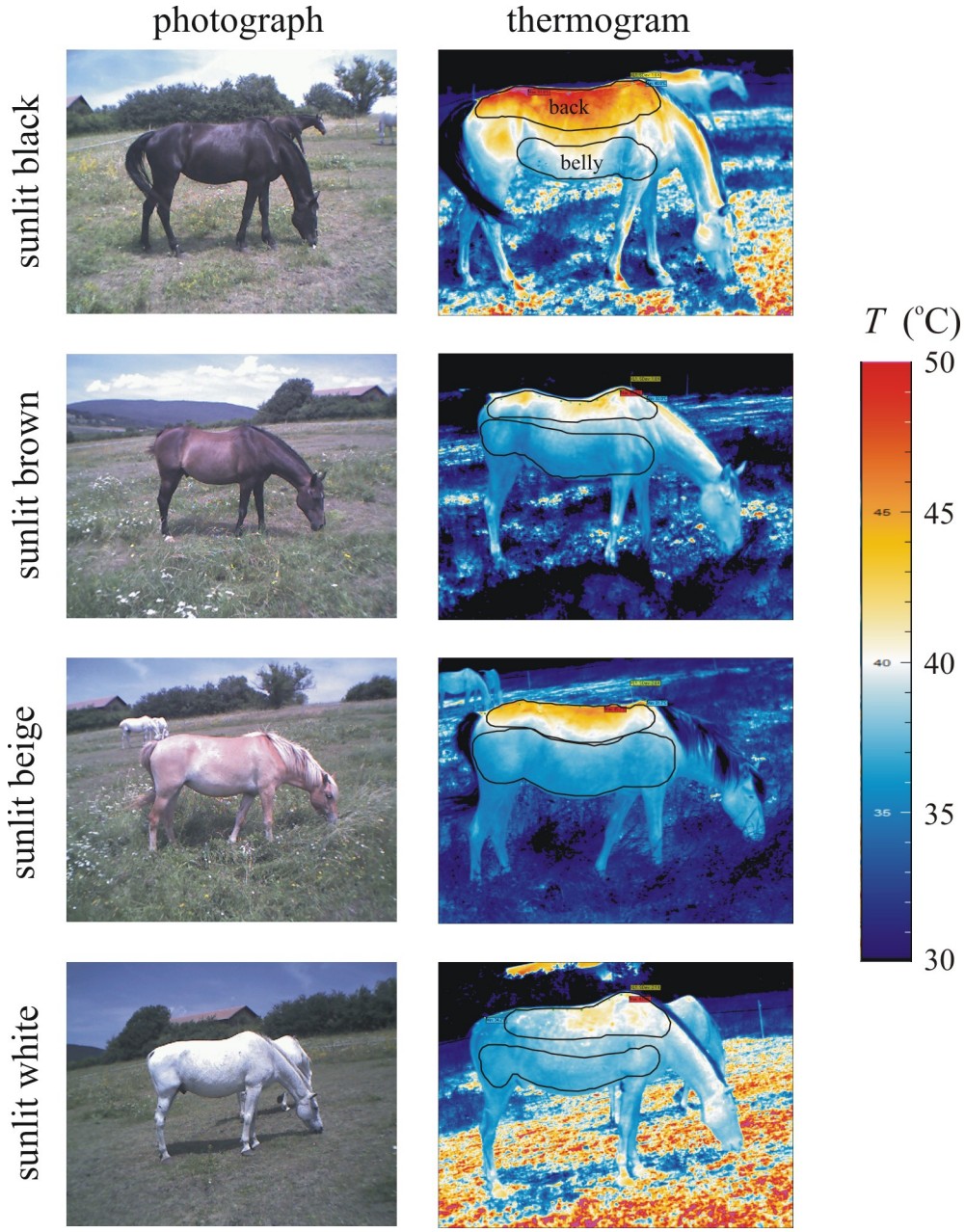

**Fig 1. Thermograms of horses.** Photographs and thermograms of sunlit black, brown, beige and white horses. In the thermograms the black perimeters of the back and belly areas are shown where the surface temperature $T$ was averaged.

As illustrated in Fig 4A, the number $N_e$ of escaped tabanids that landed on barrels has a maximum at around 41˚C, drops to zero at 17˚C and decreases almost to zero at 62˚C. The drop of $N_e$ at lower surface temperatures was the result of (i) less tabanids landing on colder surfaces, and (ii) the escape success is lower on them (see Fig 4C). Since the landing events shorter than 10 seconds were not registered, $N_e$ droped with increasing $T$. In Fig 4B, the number $N_c$ of captured tabanids that landed on barrels exhibits a clear decreasing trend with increasing $T_{barrel}$ for both temperature intervals of $17˚C \leq T \leq 62˚C$ and $T_{min} = 31˚C \leq T \leq T_{max,BL} = 55˚C$. In

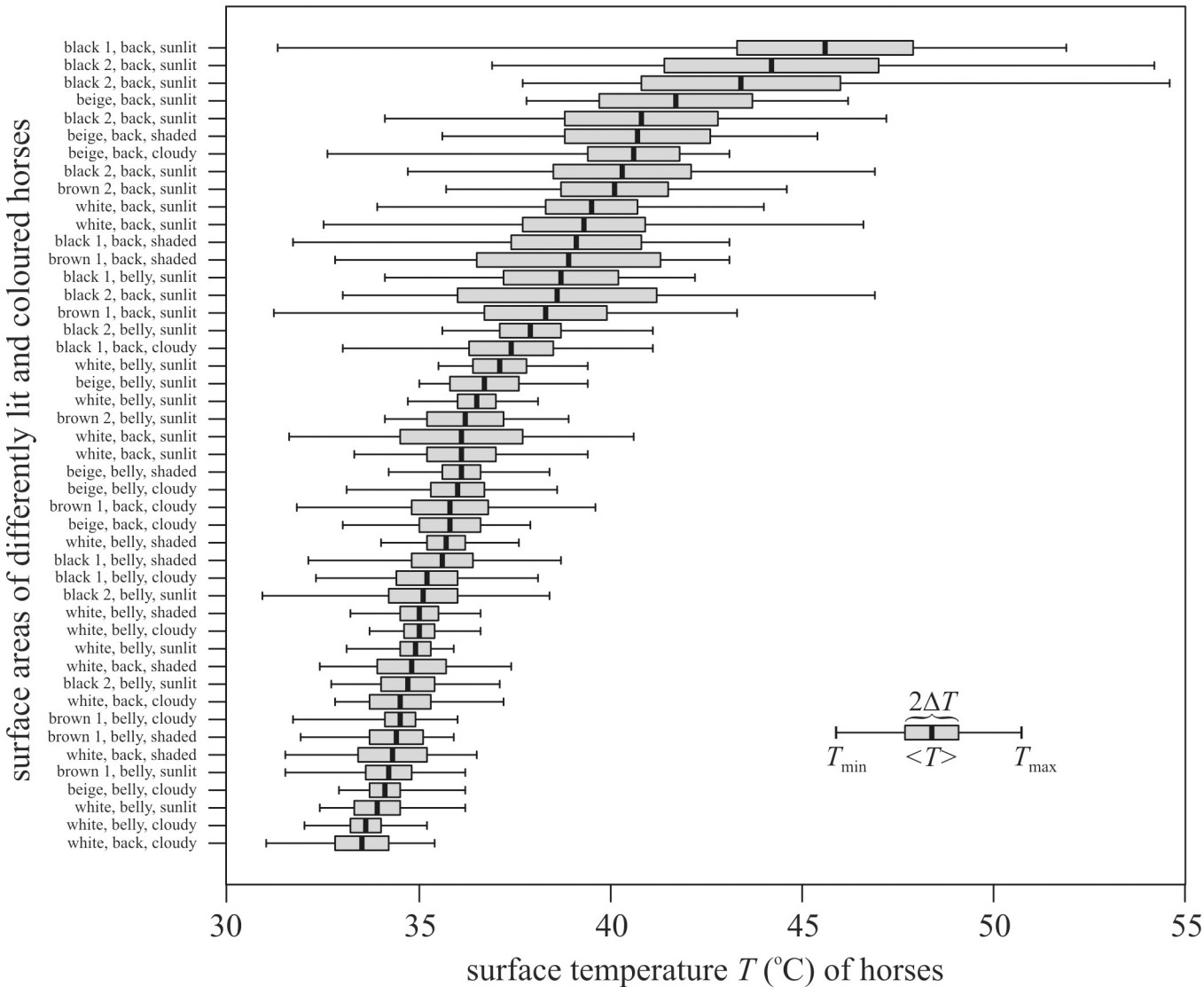

**Fig 2. Surface temperatures of horses.** Minimum ($T_{min}$), maximum ($T_{max}$), average ($<T>$) and standard deviation ($\Delta T$) of the surface temperature of the back and belly of black, brown, beige and white horses measured with thermography under different illumination conditions (S1–S4 Tables, S1–S4 Figs). shaded: shaded side of a sunlit horse, sunlit: sunlit side of a sunlit horse, cloudy: the horse was illuminated by skylight when the sun was occluded by clouds.

Fig 4C the increasing tendency of the normalized escape succes $e = N_e/(N_e + N_c)$ with increasing barrel surface temperature $T_{barrel}$ is clear. Fig 4C also illustrates that the systematic increase of the escape success $e$ is also present in the temperature range $T_{min} = 31°C \leq T \leq T_{max,BL} = 55°C$ that is typical for the surface temperature of horses.

Fig 5A shows that $T_{barrel}$ and $T_{air}$ correlate positively. Similarly, there was a positive correlation between the normalized escape success $e = N_e/(N_e + N_c)$ and $T_{air}$ if we take into consideration the results of all five experiments (Fig 5B). Since $T_{air}$ and $T_{barrel}$ correlate positively (Fig 5A) and $e$ increases with increasing $T_{barrel}$ (Fig 4C), it could also be expected that $e$ increases with increasing $T_{air}$ as seen in Fig 5B. However, applying a linear regression for $e$-values measured at air temperatures lower than 33°C only (this way the warmest observations are eliminated when only sunlit air-filled warm barrels were used in experiment 1 resulting in a strong bias in the escape success $e$), the regression line becomes horizontal (Fig 5C). In this case there

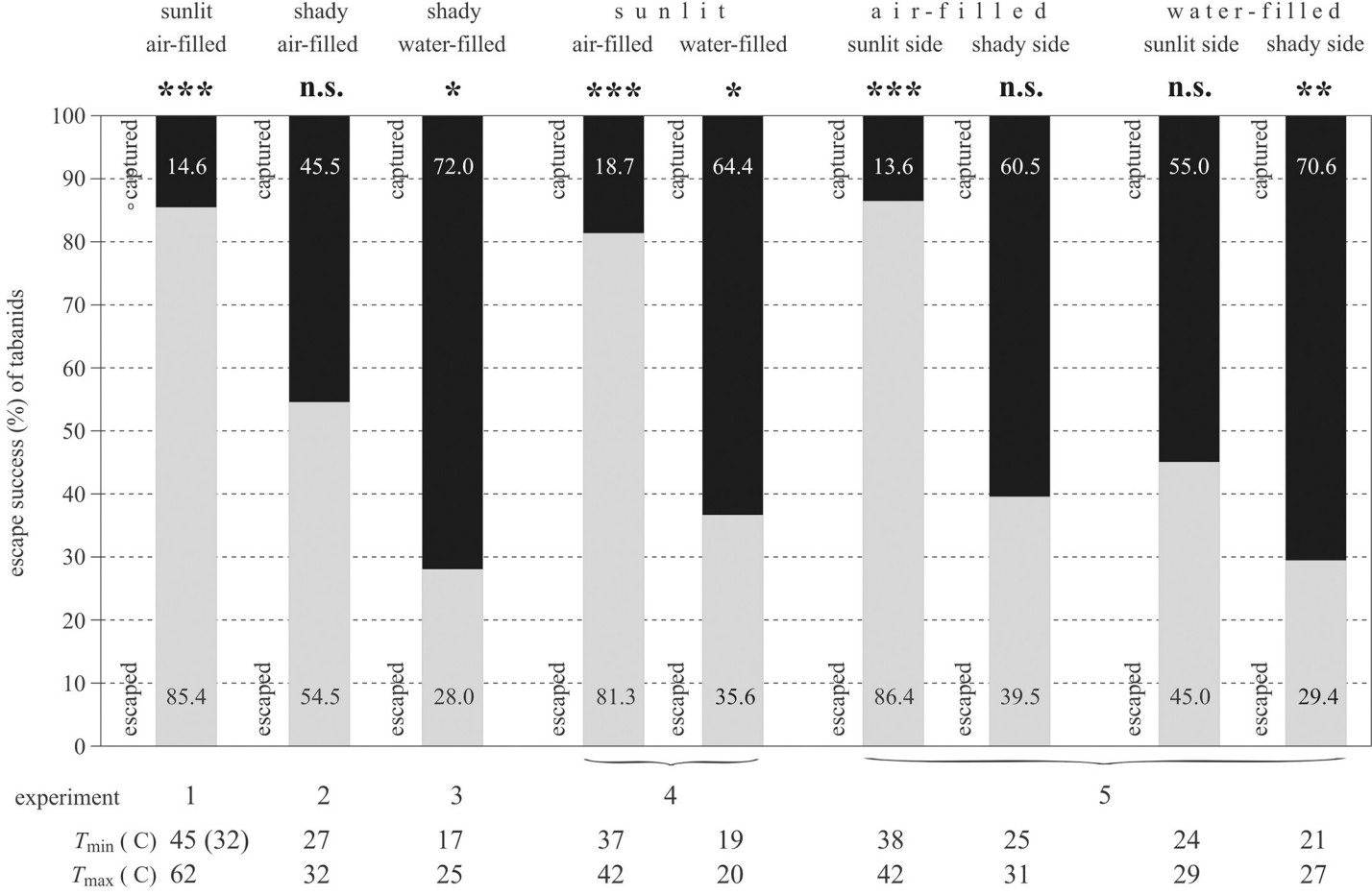

**Fig 3. Escape success of horseflies in experiments 1–5.** Surface temperature range $T_{min} \leq T \leq T_{max}$ of barrels and the proportion of escape success of tabanids (grey bars) that landed on the barrels under different illumination and thermal conditions in experiments 1–5 (S6–S10 Tables). The results of $\chi^2$ test are indicated: n.s: not significant, p > 0.05, *: 0.01 < p < 0.05, **: 0.001 < p < 0.01, ***: p < 0.001 (S5 Table). Grey and black bars illustrate the escape and capture rates, the proportion values of which are given in the columns.

is no correlation between $e$ and $T_{air}$. This suggests that the descended horseflies spent sufficient time (10 seconds) on the barrel so that $T_{barrel}$ determined the escape success, rather than $T_{air}$.

Fig 5D illustrates that the normalized escape success $e$ positively correlates with the temperature difference $\Delta T = T_{barrel} - T_{air}$ (°C) for all five experiments. Note that the 17–62°C range of $T_{barrel}$ is larger than the 23–38°C range of $T_{air}$. This means that if $\Delta T$ is low/high, then $T_{barrel}$ is also low/high. Thus, the result in Fig 5D is similar to that in Fig 4C, because only the temperature range (horizontal axis) was changed, which resulted in some blur due to the relatively small variation $T_{air}$.

Fig 6A shows the results of a logistic regression show a highly significant ($p < 0.0001$) positive correlation between $T_{barrel}$ and the escape probability $\varepsilon$ of descended tabanids (S11 Table). The effect of $T_{barrel}$ in the logistic regression was also significant ($p = 0.000395$) for the temperature range 31–55°C (S12 Table). The logistic regression in Fig 6B displays a positive correlation between the air temperature $T_{air}$ and the escape probability $\varepsilon$ of tabanids, and the effect of $T_{air}$ was highly significant ($p < 0.0001$, S13 Table). Fig 6C shows the highly significant ($p < 0.0001$) positive correlation between the difference $T_{barrel} - T_{air}$ and the escape probability $\varepsilon$ of tabanids (S14 Table). These findings correspond to the results of the linear regressions.

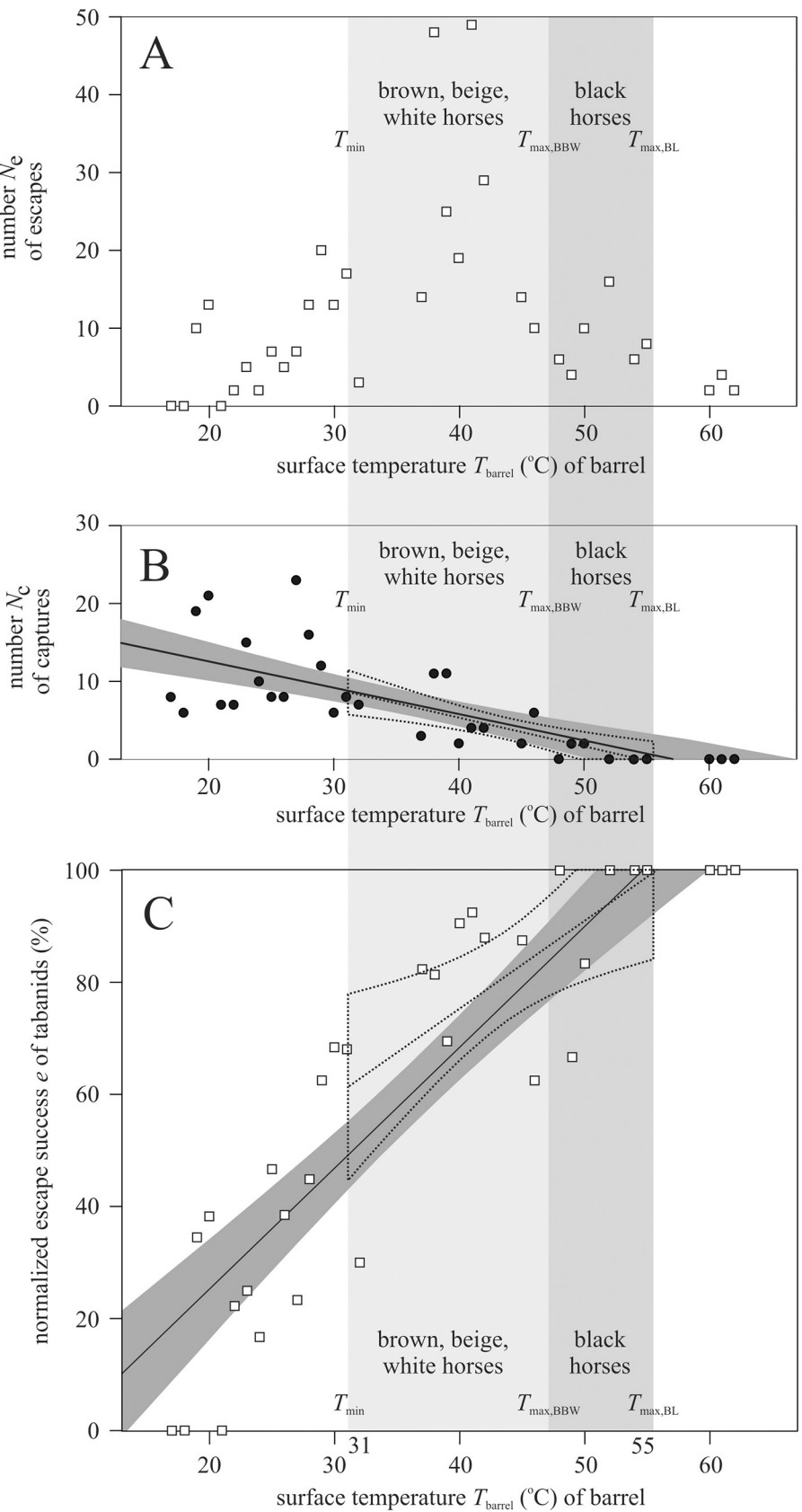

**Fig 4. Escape success of horseflies versus surface temperature.** Number $N_e$ of escaped (A) and number $N_c$ of captured (B) tabanids that landed on barrels as a function of the surface temperature $T_{barrel}$ ($^{\circ}$C). In B a continuous straight line indicates the linear fit to all $N_c(T_{barrel})$ data (black circles in the interval $17^{\circ}C \leq T \leq 62^{\circ}C$), and a dark grey band around this fit shows the 95% confidence interval. (C) Normalized escape success $e = N_e/(N_e + N_c)$ versus $T_{barrel}$ ($^{\circ}$C). A continuous straight line and a dark grey band illustrate the linear fit to all $e(T_{barrel})$ data (white squares in the interval $17^{\circ}C \leq T \leq 62^{\circ}C$) with 95% confidence interval. The vertical light and medium grey columns denote the interval $T_{min} = 31^{\circ}C \leq T \leq T_{max,BBW} = 47^{\circ}C$ of the surface temperature of brown, beige and white horses (BBW) and the interval $T_{min} = 31^{\circ}C \leq T \leq T_{max,BL} = 55^{\circ}C$ of black (BL) horses measured by thermography (Fig 2). In B and C a dotted straight line and a 95% confidence interval with dotted perimeter illustrate the linear fit to the data within the $31^{\circ}C \leq T \leq 55^{\circ}C$ interval.

## Discussion

Female tabanids prefer to attack sunlit against shaded dark host animals, and dark against bright hosts for a blood meal, the exact reasons for which were unknown. Our results presented here show that the surface temperature of sunlit darker horses is higher than that of sunlit brighter horses. This result corresponds to previous measurements [21, 22, 23, 24, 25, 26, 27]. The differences in surface temperatures of dark and bright as well as sunlit and shaded hosts may partly explain their different attractiveness to tabanids. Horváth *et al.* [20] found that *Tabanus tergestinus* horseflies prefer sunlit warm shiny black targets against sunlit or shaded cold ones with the same optical characteristics. They hypothesized that blood-sucking female tabanids prefer higher temperatures, because their wing muscles are quicker and their nervous system functions better in a warmer microclimate [28], therefore they can avoid the parasite-repelling reactions of host animals by prompt takeoffs. Since the thermoreceptors of tabanids (as in Diptera in general) are in their legs, antennae and mouthpart [28, 29, 30], they cannot sense (e.g. by infrareceptors) the temperature of a target remotely. They can sense the surface temperature of a substrate/host only after physical contact (landing). However, based on the leg/antenna/mouth-sensed temperature of the boundary layer around a target, tabanids can decide whether the target's surface is or is not warm enough for alighting [20].

The blood meal from warm-blooded animals is used by biting female horseflies as an energy source for the maturation of their eggs [25, 28, 29, 30]. For this purpose, the blood of any warm-blooded host is sufficient, regardless of whether a host is dark- or bright-coloured, shaded or sunlit. In spite of this, blood-seeking female tabanids prefer dark and sunlit hosts [1, 2, 3, 4, 5, 6, 7], and this is the reason why horsefly traps usually have black decoys and are most effective in sunshine [8, 9, 10, 11, 12, 13, 14, 15, 16]. Our main assumption was that blood-seeking tabanids prefer to land on sunlit dark hosts to keep their body warm, which aids their rapid escape when the host performs such typical antiparasite reactions as removing horseflies from their coats with tail brushing, stamping and dislodging the flies, or by nibbling their skin [1, 28]. These fly-repelling reactions are dangerous for blood-sucking tabanids, therefore have to be avoided by a quick flying away.

In this work we analysed the results of our field experiments for the whole tabanid population of the study area without considering differences between tabanid species/genus, because species/genus identification was not feasible in the field. However, there may be differences between species/genus in landing, daily activity and responses to environmental parameters that might influence their escape success. For example, *Haematopota* species might be more active in the late afternoon when $T_{air}$ decreases. It is also unknown whether the influence of $T_{air}$ on host preference for *Haematopota* sp. is lower than that for *Tabanus* or *Atylotus* sp. To test these hypotheses could be the focus of further studies. What we know from our field experiments is the following: (i) apart from $T_{air}$ (S6–S10 Tables) the weather situation (calm with no meteorological fronts) was the same during our field experiments. (ii) There was no

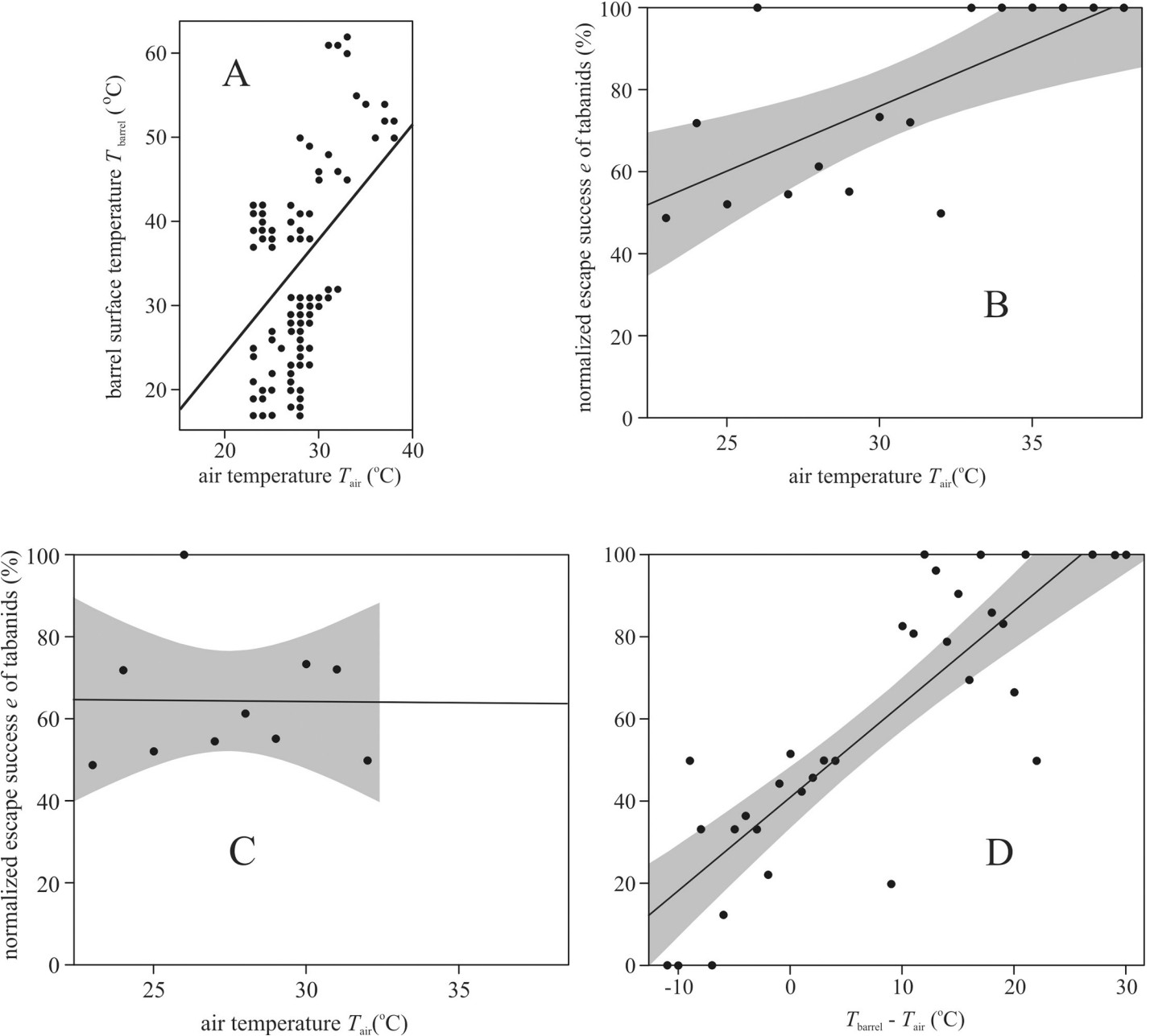

**Fig 5. Normalized escape success versus temperatures.** (A) The barrel surface temperature $T_{barrel}$ (°C) versus air temperature $T_{air}$ (°C) in field experiments 1–5. A straight line indicates the linear fit to the $T_{barrel}$ ($T_{air}$) data (black circles). (B) Normalized escape success $e$ versus $T_{air}$ (°C) for all five experiments. (C) As B, but only for $T_{air} \leq 32°C$. (D) Normalized escape success $e$ versus $T_{barrel}$—$T_{air}$ (°C) for all five experiments. In B-D continuous straight lines indicate the linear fit to the data (black circles), and dark grey bands around the fit show the 95% confidence intervals.

correlation between the escape success $e$ of descended horseflies and $T_{air} < 33°C$ (Fig 5C). In this case $T_{barrel}$ determined the escape success $e$, rather than $T_{air}$.

According to earlier field experiments [31, 32] in the same experimental site (a Hungarian horse farm in Szokolya) with the same tabanid species (*Tabanus tergestinus, T. bromius, T. bovinus, T. autumnalis, Atylotus fulvus, A. loewianus, A. rusticus, Haematopota italica*) as in the present field experiments, the daily activity of different tabanid species and the effect of

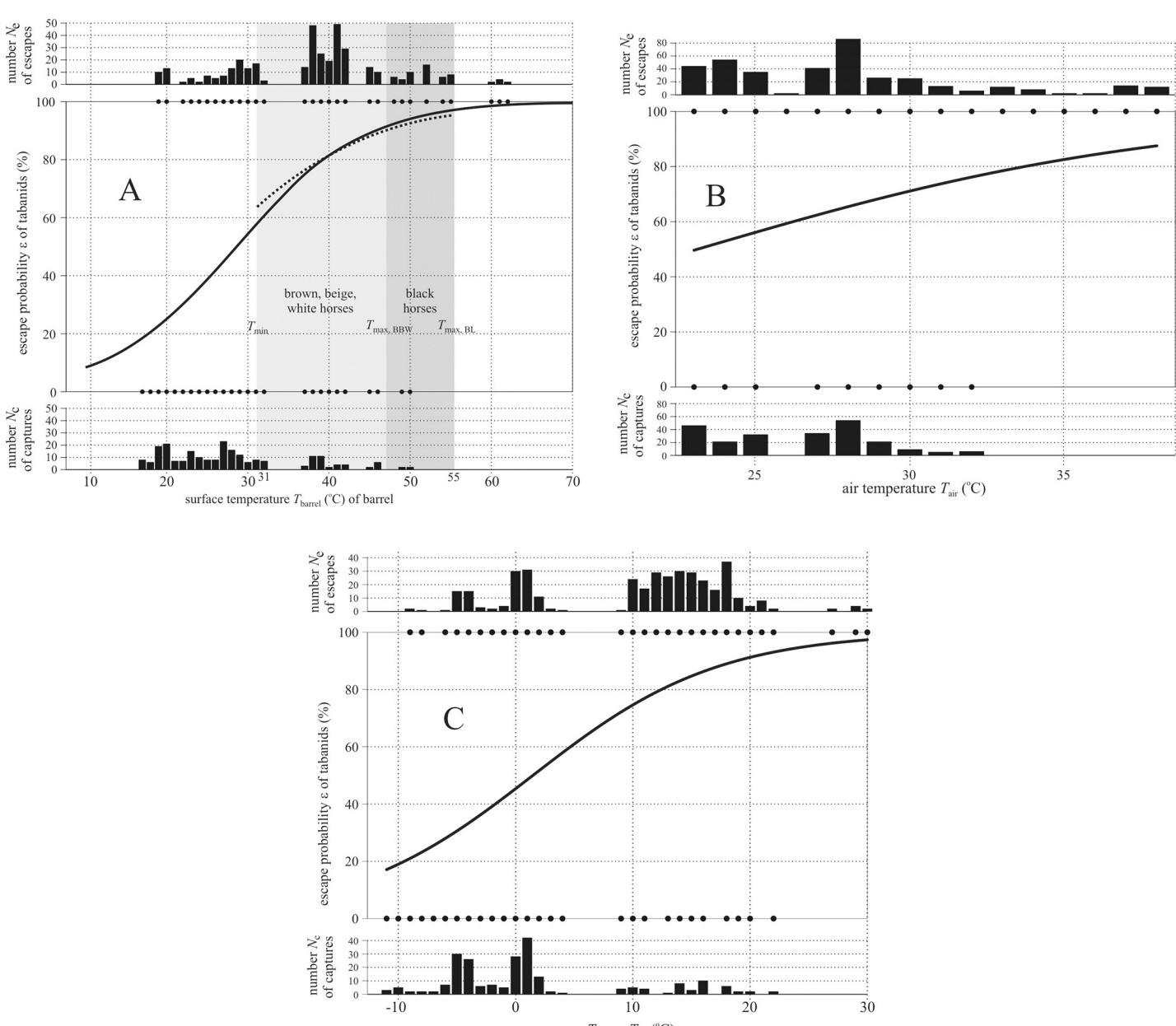

**Fig 6. Results of logistic regressions.** (A) *Top*: Number $N_e$ of escaped horseflies versus the barrel surface temperature $T_{barrel}$. *Middle*: Escape probability $\varepsilon$ of horseflies, where the logistic curve is fitted to the dots showing the surface temperatures at which tabanids were escaped ($\varepsilon$ = 100%) and captured ($\varepsilon$ = 0%) within the 17˚C $\leq T_{barrel} \leq$ 62˚C interval. A dotted curve illustrates the logistic fit to the data within the 31˚C $\leq T_{barrel} \leq$ 55˚C interval. *Bottom*: Number $N_c$ of captured horseflies versus $T_{barrel}$. The vertical light and medium grey columns denote the interval $T_{min}$ = 31˚C $\leq T \leq T_{max,BBW}$ = 47˚C of the surface temperature of brown, beige and white (BBW) horses and the interval $T_{min}$ = 31˚C $\leq T \leq T_{max,BL}$ = 55˚C of black (BL) horses measured by thermography (Fig 2). (B) As A versus the air temperature $T_{air}$. (C) As A versus the difference $T_{barrel}$—$T_{air}$.

weather variables on their flight activity were slightly different. Herczeg *et al.* [32], for example, found the following: (i) rainfall, air temperature, and sunshine were the three most important factors influencing the number of tabanids trapped. (ii) The effect of relative air humidity $H$ on tabanids was indirect through the $T_{air}$: $H \approx$ 35% (corresponding to $T_{air} \approx$ 32˚C) was optimal for tabanid capture, and tabanids were not captured at $H \geq$ 80% (corresponding to $T_{air} \leq$

18˚C). (iii) A fast decrease in the air pressure enhanced the trap success for horseflies. (iv) Wind velocities exceeding 10 km/h drastically reduced the number of trapped tabanids.

In our field experiments 4 and 5 warm and cold sunlit barrels were used simultaneously, while in experiments 1–3 the cold and the warm barrels were tested separately (experiment 1: sunlit air-filled barrels, experiment 2: shaded air-filled barrels, experiment 3: shaded water-filled barrels). This was, however, not a problem, because apart from the air temperature (S6–S10 Tables) the environmental conditions (calm with no meteorological fronts) were practically identical on all experimental days. Thus, the slightly different environmental factors in our field experiments could have resulted in only small differences in the activity of a given tabanid species.

In our field experiments the investigator waited 10 seconds before he tried to capture a tabanid that landed on a barrel. This 10-second period turned out to be optimal: it was neither too short, nor too long. Within a period shorter than 10 s the flight muscles of descended tabanids could not warm up or cool down to the surface temperature of barrels [33]. On the other hand, when a tabanid recognizes that the barrel is not a host animal after landing, it flies away after a certain period. If too much time had elapsed after tabanids landed before the counting was initiated, many of these events would have been missed. Since $T_{air}$ did not correlate with the escape success (Fig 5C), 10 s was sufficiently long to warm the wing muscles of flies that landed on the barrel. On the other hand, if a tabanid lands on a sunlit dark (warm) surface, then it will not cool down, contrary to a bright (colder) surface where its wing muscles can cool down. It is reasonable to suppose that during flight the wing muscles are appropriately warm. Furthermore, according to Heinrich [33], many Dipteran species (including true flies and also tsetse flies) can be heterothermic, or generate a certain amount of heat which can then be used to improve their performance. However, as far as we know, practically nothing is known about the thermoregulation in horseflies. Thus, without knowing the horsefly thermal physiology, we restricted our study to the correlation between (air/barrel) temperature and escape success with a waiting period of 10 seconds before the experimenter tried to capture a tabanid that landed on a barrel.

During the field experiments all barrels were optically uniformly attractive to host-seeking horseflies. Our air-filled sunlit warm barrels (37–62˚C) thermally imitated sunlit black horses (31–55˚C), while our water-filled sunlit cold barrels (19–29˚C) were cooler than sunlit brown, beige and white horses (31–47˚C). Although the measured low escape successes in the case of our water-filled cold barrels were associated with temperatures that were lower than those on the studied horse bodies, the positive correlation between escape success $e$ and surface temperature $T$ (i.e. increasing $e$ with increasing $T$) is also evident in the interval $T_{min} = 31˚C \leq T \leq T_{max,BL} = 55˚C$ as clearly shown by the dotted curves of Figs 4C and 6A. Thus, the results of our field experiments show that the escape success of tabanids depends on the host's surface temperature $T$: the higher the $T$, the larger the escape success of horseflies. The average surface temperature of the studied sunlit black horses was between 48 and 55˚C. Almost all tabanids managed to escape in this temperature range.

In Figs 4 and 6A, the 100% escape successes at surface temperatures $T > 50˚C$ are associated with very small absolute numbers of horseflies and also with hot air temperatures ranging from 33 to 38˚C (S6 Table). The reason for this is that only the minority of tabanids landing on hot ($T > 55˚C$) surfaces spent periods longer than 2 seconds on them, and the waiting time until a catch attempt was 10 seconds after a tabanid landed on a barrel. In experiment 1 the weather was very warm ($T_{air} \leq 38˚C$) and the surface temperature of the warm barrel was above 45˚C the whole day (S6 Table). Larger horseflies can tolerate extreme temperatures for a short time, and are also intrinsically faster [28].

The thermal transfer between a horsefly and a surface could be very different in the following two cases: (i) When a fly lands on a horse, the grip is maintained by the insect's legs holding on the hairs, leaving plenty of insulating air between the two animals. (ii) When a fly lands on a plastic barrel, its legs must remain attached to a much less convenient substrate. In the future, an important task could be to measure both types of thermal transfer.

It has been shown that dark-bodied host animals have a much stronger reflection-polarization signature than bright-bodied ones, which is an important visual sign, leading to horsefly attacks [18]. The high degrees of polarization of reflected light helps tabanids to select sunlit dark host animals from the dark patches of their visual environment. A strong polarization signature could also advertise a hot animal and greater chances of escape of tabanids, but both traits could be also dissociated, all depending on the thermal transfer and the thermal physiology of horseflies.

## Conclusion

We presented the results of our field experiment studying the dependence of tabanid escape success on the temperature of the landing surface. The temperature of artificial landing sites was parallelled by the analysis of thermal imaging measurements of surface temperatures of different body parts of differently coloured horses. Not surprisingly, fur temperatures were higher in darker horses and lower in bright coloured ones. The tabanid escape success strongly depended on the surface temperature; the highest escape success occurred on surfaces having temperatures similar to those recorded in black horses, i.e. above 50°C. We conclude that the warmer (also darker) host animals allow higher escape success of blood-sucking horseflies. This supports our hypothesis that the preference of horseflies to dark hosts has partly evolved due to higher survival success.

## Materials and methods

### Animal ethics statement and field study permits

Csaba Viski permitted us to photograph his horses in his horse farm in Szokolya. For the location and activities of our field study no specific permissions were required.

### Thermography of horses

On a warm day (6 July 2019) 46 thermograms of 2 black, 2 brown, 2 beige and 2 white horses were obtained with an infrared camera (VarioCAM®, Jenoptik Laser Optik Systeme GmbH, Jena, Germany, nominal precision of ±1.5°C) under sunny and cloudy conditions. The validation and calibration of this thermocamera with a contact thermometer (GAO Digital Multitester EM392B 06554H, EverFlourish Europe Gmbh., Friedrichsthal, Germany, nominal precision of ±1°C) are described in the Supporting Material of [20].

### Field experiments

**Field experiments 1–5** were performed on 1, 2, 3, 4 and 11 July 2019 on a Hungarian horse farm in Szokolya (47° 52' North, 19° 00' East), where horseflies were present. All five experimental days were windless, and only weak local winds blew in the early afternoons. Meteorological fronts did not move through the study site. Thus, apart from the air temperature (according to S6–S10 Tables, in experiment 1 the air temperatures were warmer than on the other four experimental days), the environmental conditions were practically the same. In the mornings, the weather was sunny, warm and cloudless, however in the afternoons a few cumulus clouds formed. In these experiments the escape success of tabanids that landed on shiny

black cylindrical plastic barrels (height = 42 cm, diameter = 30 cm, wall thickness = 5 mm) of different surface temperatures (set with warm air or cold water load) but with the same optical characteristics was studied under sunlit and shaded conditions. The purpose of these barrels was to imitate warm and cold dark host animals of tabanids. Sufficiently large temperature differences between the warm and cold barrels could easily be ensured with air-filled warm and water-filled cold barrels. An experimenter, who was "blind" to the predictions of the experiment tried to capture the tabanids that landed on the barrels with a hemispherical tea-strainer of diameter 15 cm. The time allowed to elapse before a capture attempt was 10 seconds (measured with a stopwatch) after a tabanid landed on a barrel. This 10-second period turned out to be optimal: it was neither too short, nor too long (a more detailed explanation of the choice of this optimal 10-second value can be read in the Discussion).

- In **experiment 1** (1 July 2019, 10:20–17:00 hour = local summer time = UTC + 2 h) two air-filled sunlit black barrels were used, which thermally modelled sunlit black host animals (e.g. horses) for tabanids. The two barrels were put on top of each other, and both were placed on a four-legged white plastic stand (height = 46 cm) at a sunlit site without any shade cast by vegetation or other objects. Only tabanids that landed on the sunlit side of the barrels were taken into account.

- In **experiment 2** (2 July 2019, 9:40–16:00 h) two air-filled barrels under shadow were used, which modelled shaded hosts. The barrels were put on top of each other and the white stand was in the shade of trees during the experiment. Only tabanids that landed on the side of the barrels facing toward the open field were considered.

- In **experiment 3** (3 July 2019, 9:50–16:00 h) two cold-water-filled shaded barrels were used, which modelled cool shaded hosts. The barrels were continuously in the shadow of trees. Both barrels were filled with tap water and 10 frozen ice packs (Aspico G40, 0.25 litre, 0.76 kg). The experimenter tried to capture only tabanids that landed on the side of the barrels facing toward the open field.

- In **experiment 4** (4 July 2019, 10:00–12:00 h) two sunlit air-filled barrels and two sunlit cold-water-filled barrels were used which thermally modelled sunlit and shaded hosts, respectively. Both barrels were continuously exposed to sunlight. Only tabanids that landed on the sunlit side of the barrels were tried to capture.

- **Experiment 5** (11 July 2019, 10:20–16:00 h) was technically the same as experiment 4, but all tabanids landing on both sunlit and shaded sides of barrels were subject of attack.

The experimenter wore white clothes and a hat against direct sunshine and to minimize his visual attractiveness to tabanids. He was sitting on a chair during the experiments next to the barrels (50 cm) in such a way that he could easily reach the tabanids on the barrels with the tea-strainer. After the fly was successfully caught, it was released. After each capture trial the air temperature ($T_{air}$) and the surface temperature of the barrel ($T_{barrel}$) at the tabanid's landing location was measured with a contact thermometer (GAO Digital Multitester EM392B 06554H, EverFlourish Europe Gmbh., Friedrichsthal, Germany, nominal precision of ±1˚C). For this part of the study, the use of thermography was not possible, because (i) the thermocamera needed about two minutes for self-calibration after each switch on, whilst the next tabanids could land on the barrels, and (ii) on the recorded thermograms it would have been impossible to localize the exact landing sites of tabanids. The experimenter was the same person throughout all experiments, who had practised the capture of tabanids during a pilot experiment. Due to the low number of flying tabanids in the vicinity of the barrels, only single

tabanids landed on the barrels at any given time. Thus, the experimenter's attention could focus entirely on one fly at a time.

An *in situ* identification of the species of tabanids that landed on the barrels was not feasible. It was obvious, however, that they were tabanids (Diptera: Tabanidae). In previous field experiments [31, 32], the following tabanid species occurred at the same study site: *Tabanus tergestinus*, *T. bromius*, *T. bovinus*, *T. autumnalis*, *Atylotus fulvus*, *A. loewianus*, *A. rusticus*, *Haematopota italica*. Since we could record the escape success of different tabanid species, our results can be considered as the average escape success of the tabanid population of the experimental site.

Since the reflection-polarization characteristics of the dry barrel surface are independent of its temperature in the visible spectral range, all optical parameters (radiance, degree of linear polarization and angle of polarization) of our warm and cool barrels were identical.

### Statistical analysis

For comparison of the numbers of escaped and captured tabanids that landed on test surfaces of various temperatures, we applied $\chi^2$ tests of homogeneity, where the escape versus non-escape ratio was tested against the predicted 50/50 ratio. These $\chi^2$ tests were performed to compare escape/non-escape numbers for a given barrel or barrel side (sunlit or shaded). Thus, the compared escape/non-escape numbers corresponded to the same barrel temperature and there was no comparison between data originating from different barrel temperatures. In other words, $\chi^2$ test was used to detect whether a given barrel temperature had an effect on tabanid escape success.

Linear regressions were applied to find a trend of the escape success of horseflies as a function of $T_{barrel}$, $T_{air}$ and $T_{barrel}$—$T_{air}$. The independent variables were $T_{barrel}$, $T_{air}$ and $T_{barrel}$—$T_{air}$, while the dependent variable was the normalized escape success $e = N_e/(N_e + N_c)$, where $N_e$ is the number of escaped tabanids and $N_c$ is the number of captured horseflies. We also applied logistic regression to model the probability of escape as a function of $T_{barrel}$, $T_{air}$ and $T_{barrel}$—$T_{air}$. We also applied the linear and logistic regressions as a funciton of $T_{barrel}$ using the data within the $31°C \leq T \leq 55°C$ interval. Logistic regression was also used to find whether there is a correlation between $T_{air}$ and $T_{barrel}$. The R statistical package 3.0.2 [34] was used for statistical analyses.

### Supporting information

**S1 Table. Temperatures of black horses measured with thermography on shaded and sunlit sides of the back and belly, and when the sun was occluded by clouds (cloudy).** $<T>$: average, $\pm\Delta T$: standard deviation, $T_{min}$: minimum, $T_{max}$: maximum.
(DOC)

**S2 Table. Temperatures of brown horses measured with thermography on shaded and sunlit sides of the back and belly, and when the sun was occluded by clouds (cloudy).** $<T>$: average, $\pm\Delta T$: standard deviation, $T_{min}$: minimum, $T_{max}$: maximum.
(DOC)

**S3 Table. Temperatures of beige horses measured with thermography on shaded and sunlit sides of the back and belly, and when the sun was occluded by clouds (cloudy).** $<T>$: average, $\pm\Delta T$: standard deviation, $T_{min}$: minimum, $T_{max}$: maximum.
(DOC)

**S4 Table. Temperatures of white horses measured with thermography on shaded and sunlit sides of the back and belly, and when the sun was occluded by clouds (cloudy).** $<T>$:

average, $\pm\Delta T$: standard deviation, $T_{min}$: minimum, $T_{max}$: maximum.
(DOC)

**S5 Table. Results of $\chi^2$ tests comparing the sums of S6–S10 Tables obtained in experiments 1–5.**
(DOC)

**S6 Table. Capture success (-: Not captured, +: Captured) of horseflies, and temperatures of the air ($T_{air}$) and the surface of the air-filled sunlit barrel ($T_{barrel}$) in experiment 1 on 1 July 2019.**
(DOC)

**S7 Table. Capture success (-: Not captured, +: Captured)) of horseflies, and temperatures of the air ($T_{air}$) and the surface of the air-filled shaded barrel ($T_{barrel}$) in experiment 2 on 2 July 2019.**
(DOC)

**S8 Table. Capture success (-: Not captured, +: Captured) of horseflies, and temperatures of the air ($T_{air}$) and the surface of the cold-water-filled shaded barrel ($T_{barrel}$) in experiment 3 on 3 July 2019.**
(DOC)

**S9 Table. Capture success (-: Not captured, +: Captured) of horseflies, and temperatures of the air ($T_{air}$) and the sunlit side of the surface of the air-filled warm barrel ($T_{warm}$) and the water-filled cold barrel ($T_{cold}$) in experiment 4 on 4 July 2019.**
(DOC)

**S10 Table. Capture success (-: Not captured, +: Captured) of horseflies, and temperatures of the air ($T_{air}$) and the surface of the air-filled sunlit barrel ($T_{warm}$) and the cold-water-filled sunlit barrel ($T_{cold}$) in experiment 5 on 11 July 2019.**
(DOC)

**S11 Table. Summary of the logistic regression.** The escape probability $\varepsilon$ of tabanids depends highly significantly on the barrel surface temperature $T_{barrel}$ in the interval $17°C \leq T_{barrel} \leq 62°C$. The large difference between the null deviance and the residual deviance suggests that the logistic regression model is accurate.
(DOC)

**S12 Table. Summary of the logistic regression.** The escape probability $\varepsilon$ of tabanids depends highly significantly on the barrel surface temperature $T_{barrel}$ in the interval $31°C \leq T_{barrel} \leq 55°C$.
(DOC)

**S13 Table. Summary of the logistic regression.** The escape probability $\varepsilon$ of tabanids depends highly significantly on the air temperature $T_{air}$.
(DOC)

**S14 Table. Summary of the logistic regression.** The escape probability $\varepsilon$ of tabanids depends highly significantly on the temperature difference $T_{barrel}$—$T_{air}$.
(DOC)

**S1 Fig. Photographs, thermograms and thermograms with selected back and belly areas of black horses under different illumination conditions.** Shaded: shaded side of the sunlit horse. sunlit: sunlit side of the sunlit horse. cloudy: illuminated by skylight when the sun was

occluded by clouds.
(DOC)

**S2 Fig. Photographs, thermograms and thermograms with selected back and belly areas of brown horses under different illumination conditions.** Shaded: shaded side of the sunlit horse. sunlit: sunlit side of the sunlit horse. cloudy: illuminated by skylight when the sun was occluded by clouds.
(DOC)

**S3 Fig. Photographs, thermograms and thermograms with selected back and belly areas of a beige horse under different illumination conditions.** Shaded: shaded side of the sunlit horse. sunlit: sunlit side of the sunlit horse. cloudy: illuminated by skylight when the sun was occluded by clouds.
(DOC)

**S4 Fig. Photographs, thermograms and thermograms with selected back and belly areas of white horses under different illumination conditions.** Shaded: shaded side of the sunlit horse. sunlit: sunlit side of the sunlit horse. cloudy: illuminated by skylight when the sun was occluded by clouds.
(DOC)

## Acknowledgments

We are grateful to Csaba Viski who permitted our field experiments on his horse farm in Szokolya. We thank for the valuable and constructive comments of three anonymous reviewers.

## Author Contributions

**Conceptualization:** Gábor Horváth.

**Data curation:** Gábor Horváth, Tímea Tóth, Imre Miklós Jánosi.

**Formal analysis:** Gábor Horváth, Ádám Pereszlényi, Ádám Egri.

**Funding acquisition:** Gábor Horváth.

**Investigation:** Gábor Horváth, Ádám Pereszlényi, Tímea Tóth.

**Methodology:** Gábor Horváth, Ádám Pereszlényi, Ádám Egri, Imre Miklós Jánosi.

**Resources:** Gábor Horváth.

**Software:** Ádám Egri, Imre Miklós Jánosi.

**Supervision:** Gábor Horváth.

**Validation:** Gábor Horváth, Ádám Pereszlényi, Ádám Egri, Imre Miklós Jánosi.

**Visualization:** Gábor Horváth, Ádám Pereszlényi, Tímea Tóth.

**Writing – original draft:** Gábor Horváth, Ádám Pereszlényi, Ádám Egri, Imre Miklós Jánosi.

**Writing – review & editing:** Gábor Horváth, Imre Miklós Jánosi.

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
