## [Decision Letter · Decision Letter 0]

16 Mar 2020

PONE-D-20-04024

Why Do Biting Horseflies Prefer Warmer Hosts? Tabanids can Escape Easier from Warmer Targets

PLOS ONE

Dear Prof. Horvath,

Thank you for submitting your manuscript to PLOS ONE. After careful consideration, we feel that it has merit but does not fully meet PLOS ONE’s publication criteria as it currently stands. Therefore, we invite you to submit a revised version of the manuscript that addresses the points raised during the review process.

The reviewers agree that this is an interesting manuscript but also express some concerns about the methodology employed that need to be addressed carefully. These include the lack of taxonomic identification at least to genus level of the tabanids observed and the varying conditions under which data for the various experiments were collected. Importantly though, thermal measurements of real hosts and the barrels used appear to be different which may question the conclusions drawn from the data. In addition, there are a number of apparent contradictions throughout the manuscript which should also be shortened. In their revisions the authors should carefully address the comments by the reviewers provided below.

We would appreciate receiving your revised manuscript by Apr 30 2020 11:59PM. To enhance the reproducibility of your results, we recommend that if applicable you deposit your laboratory protocols in protocols.io, where a protocol can be assigned its own identifier (DOI) such that it can be cited independently in the future. For instructions see: http://journals.plos.org/plosone/s/submission-guidelines#loc-laboratory-protocols

We look forward to receiving your revised manuscript.

Kind regards,

Heike Lutermann, PhD

Academic Editor

PLOS ONE

Journal Requirements:

3. Your ethics statement must appear in the Methods section of your manuscript. If your ethics statement is written in any section besides the Methods, please move it to the Methods section and delete it from any other section. Please also ensure that your ethics statement is included in your manuscript, as the ethics section of your online submission will not be published alongside your manuscript.

Reviewers' comments:

Reviewer's Responses to Questions

**Comments to the Author**

1. Is the manuscript technically sound, and do the data support the conclusions?

Reviewer #1: Yes

Reviewer #2: Yes

Reviewer #3: Partly

2. Has the statistical analysis been performed appropriately and rigorously? 

Reviewer #1: Yes

Reviewer #2: Yes

Reviewer #3: Yes

3. Have the authors made all data underlying the findings in their manuscript fully available?

Reviewer #1: Yes

Reviewer #2: Yes

Reviewer #3: Yes

4. Is the manuscript presented in an intelligible fashion and written in standard English?

Reviewer #1: Yes

Reviewer #2: Yes

Reviewer #3: Yes

5. Review Comments to the Author

Reviewer #1: This is a very interesting and important manuscript. Manuscript comprises important data which are well represented, illustrated and commented. However, I have one comment in chapter Introduction; last part of last sentence line 82 to 85 is not clear “though the blood of shady or bright (cooler) hosts would also good for the development of tabanid eggs”. I would like to suggest delete this in the last sentence. Blood meal of warm blooded animals biting female horseflies use as source of energy for egg maturation, regardless if hosts are dark or bright colour or in shady place or in sunlit. Everything else is excellent (abstract, introduction, materials and methods, results, discussion, conclusions and statistical analyses as well as English language).

Reviewer #2: Dear Editor,

The article is about tabanid preferences to darker hosts. The authors tested the hypothesis that tabanids prefer dark sunlit warm hosts because high temperatures might increase wing muscle and nervous system performance so that tabanid females escape easily and quickly in case of host defensive reactions. First, the authors estimated surface temperatures of horses with different coat colour. Then, they compared tabanid escape success in the field using different barrels modelling horse surface temperatures.

This study provides an interesting approach to explain why tabanid are attracted by warm hosts or targets since it is well-known that host attractivity is dependant on olfactive, visual and thermal cues. However, here are some minor concerns about the study:

- The authors analysed the results for all the tabanid population of the study area without considering differences between species. I understand that species identification is not feasible in the field, however the authors could have identified tabanid at the genus level. Tabanus, Atylotus and Haematopota have main characteristics that allow to discriminate them. There are differences between species in landing, daily activity and responses to environmental parameters that might influence their escape success. For example, Haematopota species might be more active in late afternoon when temperatures decrease. Maybe, the influence of temperatures on host preference for Haematopota sp. is lower than for Tabanus sp. or Atylotus sp. I think that this point should be discussed by the authors.

- Field experiment setting: only the experiments 4 and 5 are well set to compare tabanid escape success between the two types of barrels modelling hosts with different temperatures in the same conditions. In experiments 1, 2 and 3, the two barrels were not tested simultaneously. They were used different days with different environmental conditions. However, factors other than temperatures (for example, wind) have an effect on tabanid activity and these other factors might also influenced escape success.

- Statistical analysis : the authors used a Khi 2 test to compare the escape/non escape ratio versus the 50/50 ratio. This test did not allow to compare warm barrel versus cold barrel.

- Discussion (lines 191-194) : I suggest to develop this part and to cite some references about tabanid landings and host defensive reactions.

- I am not English-native speaker, but I think that the manuscript might be improved by an English speaker review.

Reviewer #3: The paper by Horváth et al. deals with the interesting hypothesis, demonstrated by the Authors, that horseflies, after landing, would prefer to walk around and feed on warmer hosts as this would allow them to be more reactive and ready to take off, escaping hosts defensive reactions. The paper is interesting for an entomologist working on horseflies, English language is very good, methods are sound and provided data strong, but there are some major concerns affecting the manuscript, to be evaluated before taking a decision regarding its publication on PLOS ONE.

The first one regards the exceedingly excessive length of the manuscript that is really redundant and rambling. Whole sections should be reduced or totally deleted. Authors spent a lot of words both in materials and methods and results dealing with very obvious statements regarding horse temperatures, that should be considered not a for its own sake section, but just something functional to experiments on horseflies. As a consequence, all the sections dealing with horse temperatures should be reduced to few lines, also considering that results are very obvious: a dark horse is warmer than a white one, a horse in sunlight is warmer than a horse in the shadow, the belly is cooler than the back, and other similar things. This is very boring for the reader. Also figures, photos and graphs on this subject should be eliminated. The whole manuscript needs to be simpler and slenderer to make it really interesting.

But, there are also some logic and methodological problems, with contradictory statements and results. There is a methodological problem, which in theory could impair a big part of the results, needing an explanation by the Authors. I hereby detail the problems I revealed during the review, problems that have to be addressed by Authors with a very thorough revision, to make the manuscript deserving publication on PLOS ONE.

1. Manuscript Length: a) Lines 93-110: reduce to 3-5 lines at maximum, as they are very obvious results, which should be only functional to the experiments on horseflies. Moreover, remove figures and graphs only dealing with horse temperatures; b) Lines 145-149: these lines should be deleted, as results of this analysis are clear enough also without this heavy introduction; c) Lines 197-201: delete these lines. Not necessary and only making the discussion heavier; d) Lines 209-218: delete these lines. They are an unnecessary and really unexplainable repetition of things already said elsewhere in the manuscript. They are Introduction, Materials and Methods, not a Discussion at all; e) Lines 265-278: the same as for point a). Reduce to 3 lines. This section is not for its own sake, but only functional to experiments on tabanids. Moreover, most part of relative results are very obvious. Again, shorten all the sections regarding horse temperatures and remove graphs, pics. They all make the paper very strenuous to read; f) Lines 290-292: not necessary. Delete; g) Lines 295-296: not necessary. Delete; h) Lines 326-329: shorten this explanation. It was not possible for technical problems. Most part of readers will be entomologists!!!; i) Lines 342-345: Delete these lines. Authors just said that from an optical point of view the barrels were the same, this is enough.

2. Methodological problem: What reported at lines 89-92 is quite strange and needs to be explained by the Authors. In fact, while warm and cold barrels ranges of temperatures don't overlap, warm and “cold horses” are highly overlapping. Moreover, while warm barrels are, for the most part, in the range of temperatures of “warm horses”, cold barrels temperatures are lower than those of “colder horses”, totally out of the range of temperature of “cold horses”. This is very strange. Standing this situation, it seems that authors used cold barrels too cold, don't reflecting a real situation and hence possibly producing wrong results. This is really very relevant. The use of so cold barrels could invalidate the whole study. Authors should explain this choice and why, in their opinion, this is not affecting the validity of the work.

3. Contradictory Statements and Needed Explanations: a) Lines 66-68: if it is true, as Authors say in another part of the manuscript, that tabanids are not able to feel temperature of an object before landing on it, what stated in these lines is in contradiction with what Authors try to demonstrate in the paper. Black decoys used for catching horseflies would work not for their temperature, but for their colour; b) Lines 129-144: the whole paragraph is very mixed up, difficult to read and understand. Furthermore, Authors don’t explain some of the findings. Why there is a drop of escaped tabanids at very high temperatures? Zero at 62°C? Why fewer tabanids landed on cold barrels if they feel temperature only after landing? Why the specification at lines 139-140? All these things are quite obscure for the reader. Authors should greatly simplify this section, explain these points and report only the relevant results; c) Line 294: Why 10 seconds? In Materials and Methods Authors should explain this choice; d) Lines 331-333: it is not clear how Authors prevented the simultaneous landing of two or more tabanids on barrels. They give it for granted, but it is not.

At Line 62, “spp.” not in italics;

At line 84 “would also good for” should be changed in “would also be good for”

In conclusion, the paper by Horváth et al. could deserve publication on PLOS ONE only after major changes regarding its length and some logic inconsistency that Authors should modify or better explain.

6. PLOS authors have the option to publish the peer review history of their article (what does this mean?). If published, this will include your full peer review and any attached files.

Reviewer #1: No

Reviewer #2: No

Reviewer #3: No

---

## [Author Response · Author response to Decision Letter 0]

31 Mar 2020

All specific reviewer and editor comments are responded in detail in the following uploaded file:

+TabanidThermalTrapping_PLoS-One-response.doc

---

## [Decision Letter · Decision Letter 1]

16 Apr 2020

PONE-D-20-04024R1

Why Do Biting Horseflies Prefer Warmer Hosts? Tabanids can Escape Easier from Warmer Targets

PLOS ONE

Dear Prof. Horvath,

Thank you for submitting your manuscript to PLOS ONE. After careful consideration, we feel that it has merit but does not fully meet PLOS ONE’s publication criteria as it currently stands. Therefore, we invite you to submit a revised version of the manuscript that addresses the points raised during the review process.

Both reviewers are satiesfied with the way the authors have addressed there previous comments but there are still a few matters that need to be addressed before this manuscript can be considered for publication in PLoS One. The authors may want to seek help with grammatical editing. Given that PLoS One does not provide any editorial service after acceptance at minimum the authors should address the grammatical concerns raised in the annotated pdf. These include

1. the replacement of the term 'shady' with 'shaded' as well as

2. the inclusion of 'that' before 'landed' on many occasions.

3. In addition, the authors should use their abbreviations (i.e. Tair, Tbarrel, ΔTbarrel-Tair) consistently after once brielfly introducing them at first mention.

Please find a few additional comments that require addressing in your revision in the annotated pdf attached.

We would appreciate receiving your revised manuscript by May 31 2020 11:59PM. To enhance the reproducibility of your results, we recommend that if applicable you deposit your laboratory protocols in protocols.io, where a protocol can be assigned its own identifier (DOI) such that it can be cited independently in the future. For instructions see: http://journals.plos.org/plosone/s/submission-guidelines#loc-laboratory-protocols

We look forward to receiving your revised manuscript.

Kind regards,

Heike Lutermann, PhD

Academic Editor

PLOS ONE

Reviewers' comments:

Reviewer's Responses to Questions

**Comments to the Author**

1. If the authors have adequately addressed your comments raised in a previous round of review and you feel that this manuscript is now acceptable for publication, you may indicate that here to bypass the “Comments to the Author” section, enter your conflict of interest statement in the “Confidential to Editor” section, and submit your "Accept" recommendation.

Reviewer #2: All comments have been addressed

Reviewer #3: All comments have been addressed

2. Is the manuscript technically sound, and do the data support the conclusions?

Reviewer #2: Yes

Reviewer #3: Yes

3. Has the statistical analysis been performed appropriately and rigorously? 

Reviewer #2: Yes

Reviewer #3: Yes

4. Have the authors made all data underlying the findings in their manuscript fully available?

Reviewer #2: Yes

Reviewer #3: Yes

5. Is the manuscript presented in an intelligible fashion and written in standard English?

Reviewer #2: Yes

Reviewer #3: Yes

6. Review Comments to the Author

Reviewer #2: Dear Editor

the authors have adressed all the comments of the three referees.

The manuscript is now suitable for publication.

Sincerely,

Reviewer #3: All my comments in review round 1 have been adequately addressed hence, in my opinion, the manuscript in this new version deserves pubblication on PLOS ONE

7. PLOS authors have the option to publish the peer review history of their article (what does this mean?). If published, this will include your full peer review and any attached files.

Reviewer #2: No

Reviewer #3: No

---

## [Author Response · Author response to Decision Letter 1]

22 Apr 2020

Our response is uploaded as a separate file.

---

## [Editor Report · Decision Letter 2]

28 Apr 2020

Why Do Biting Horseflies Prefer Warmer Hosts? Tabanids can Escape Easier from Warmer Targets

PONE-D-20-04024R2

Dear Dr. Horvath,

We are pleased to inform you that your manuscript has been judged scientifically suitable for publication and will be formally accepted for publication once it complies with all outstanding technical requirements.

With kind regards,

Heike Lutermann, PhD

Academic Editor

PLOS ONE
---

## [Editor Report · Acceptance letter]

30 Apr 2020

PONE-D-20-04024R2 

Why Do Biting Horseflies Prefer Warmer Hosts? Tabanids can Escape Easier from Warmer Targets 

Dear Dr. Horváth:

I am pleased to inform you that your manuscript has been deemed suitable for publication in PLOS ONE. Congratulations! Your manuscript is now with our production department. 

With kind regards,

on behalf of

Dr Heike Lutermann 

Academic Editor

PLOS ONE